# Effectiveness of community health workers involvement in smoking cessation programme: A systematic review

**Siti Hafizah Zulkiply****, Lina Farhana Ramli, Zul Aizat Mohamad Fisal, Bushra Tabassum, Rosliza Abdul Manaf** *

Faculty of Medicine and Health Sciences, University Putra Malaysia, Seri Kembangan, Malaysia

* rosliza_abmanaf@upm.edu.my

**Data Availability Statement:** All relevant data are within the manuscript and its Supporting information files.

**Funding:** Authors received no specific funding.

## Abstract

### Background

Sustainable Development Goals (SDG) has set the target to reduce premature mortalities from non-communicable diseases (NCDs) by one-third. One of the ways to achieve this is through strengthening the countries' implementation of the World Health Organization Framework Convention on Tobacco Control (WHO FCTC). Community health workers (CHWs) involvement has shown promising results in the prevention of NCDs. This systematic review is aimed at critically evaluating the available evidence on the effectiveness of involving CHWs in smoking cessation.

### Materials and methods

We systemically searched PubMed and CENTRAL up to September 2019. We searched for published interventional studies on smoking cessation interventions using the usual care that complemented with CHWs as compared to the usual or standard care alone. Our primary outcome was abstinence of smoking. Two reviewers independently extracted data and assessed study risks of bias.

### Result

We identified 2794 articles, of which only five studies were included. A total of 3513 smokers with 41 CHWs were included in the studies. The intervention duration range from 6 weeks to 30 months. The studies used behavioral intervention or a combination of behavioral intervention and pharmacological treatment. Overall, the smoking cessation intervention that incorporated involvement of CHWs had higher smoking cessation rates [OR 1.95, 95% CI (1.35, 2.83)]. Significant smoking cessation rates were seen in two studies.

### Conclusion

Higher smoking cessation rates were seen in the interventions that combined the usual care with interventions by CHWs as compared to the usual care alone. However, there were

**Competing interests:** We declared no competing interests.

insufficient studies to prove the effectiveness. In addition, there was high heterogeneity in terms of interventions and participants in the current studies.

## 1. Introduction

Non-communicable diseases (NCD) are the major cause of mortalities worldwide. Smoking is an important risk factor for the development of NCD, including cancers, and cardiovascular and respiratory diseases [1]. The United Nation General Assembly has developed an agenda for the Sustainable Developmental Goals (SDG) in 2015, containing a total of 17 Goals that all Member States have agreed to achieve by 2030 [2]. SDG 3, with the goal to "ensure healthy lives and promote well-being for all ages" includes target 3.4, which is to reduce premature mortality from NCD by one-third, and target 3a, which is to strengthen a country's implementation of the World Health Organization Framework Convention on Tobacco Control (WHO FCTC) [2,3].

In September 2005, Malaysia had participated with WHO FCTC and agreed to fulfil the demand to develop and disseminate a standard national guideline on tobacco [4]. In response, government of Malaysia (GoM) had set up a target to relatively reduce 30% of national smoking prevalence by 2025 within the National Strategic Plan for NCD (NSP-NCD). In addition, the World Health Organization (WHO) has also developed MPOWER that include measures for demand reduction: i) Monitoring tobacco use and prevention policies; ii) Protecting people from tobacco smoke; iii) Offering help to quit tobacco use; iv) Warning about the dangers of tobacco; v) Enforcing bans on tobacco advertising, promotion and sponsorship; and vi) Raising taxes on tobacco [5].

Tobacco has become a major public health enemy, accountable for more than 7 million preventable deaths yearly, and it has been forecasted that more than 8 million people will die from diseases related to tobacco use by year 2030 if pattern of smoking continues [6]. In Malaysia alone, it is estimated that more than 20,000 have died due to tobacco each year [7]. There are 933.1 million smokers worldwide in 2015 and around 80% of them live in low and middle-income countries (LMICs) [8]. According to the National Health and Morbidity Survey (NHMS) 2015, the prevalence of smokers in Malaysia was 22.8%, therefore, it is estimated that nearly five million Malaysians aged 15 years and above are smokers [4]. The mean age of smoking initiation is 18.3 years [9], with the highest prevalence of smokers being among the 25 to 44-years-old age group (28%), followed by the 45 to 64-years-old age group (20%) [4].

In many countries, smoking cessation programs (identification of smokers, advising and offering support to quit) are taking place in primary care settings [10]. Evidence for the effectiveness of these interventions in this setting administered by professional healthcare providers is well established [11]. Since 2004, Malaysia has set up quit-smoking clinics available at most primary health clinics [12]. Unfortunately, studies have shown that high mortalities in low socioeconomic areas, especially secondary to NCD are due to barriers in healthcare services [13]. Since the low socioeconomic status is an important determinant of smoking [14], it is pertinent that interventions on smoking should focus on these populations. As reported by Global Adult Tobacco Survey (GATS) 2011, the prevalence of smokers in rural areas was higher (24.3%, 95% CI 22.0, 26.7) as compared to urban area (22.7%, 95% CI 20.2, 25.4) [15]. The WHO has developed an effective community-based strategy to reduce the gap in healthcare in low-socioeconomic areas. However, the current smoking cessation services are not user-friendly and poorly understood [16].

In line with the Alma Ata declaration of the Primary Healthcare Concept, all community health strategy designs must address the community needs at the local level, be led by the community members themselves, and require involvement of communities to mobilize local resources. Task shifting, an idea conceived by the WHO, is defined as redistributing primary care responsibilities from physician to non-physician providers [17]. Task shifting can be further extended to include health workers without formal healthcare training, known as community health workers (CHWs) [18]. In 1989, a WHO study group had established a widely accepted definition of CHWs: the workers should be among the communities where they practice, be chosen by the communities, be able to answer to the public for their programs, be supported by the health system, and have less duration of training than professional workers [19]. CHWs have the potential to achieve primary health care goals, by enhancing access to care and promote a proper use of health resources through the provision of cultural and outreach links between communities and health systems. CHWs also have the possibility in reducing healthcare cost by giving health education, screening services, basic emergency services, continuum of care and client protection [19]. Furthermore, it is imperative to enhance civic engagement and ensure accountability in order to achieve the SDG goals [20,21]. There are considerable evidences supporting positive impacts of CHWs on the health of diverse populations especially on the maternal and child health, malaria and tuberculosis. Numerous studies have also evaluated the role of community interventions on major cardiovascular events and risk factors. Community-based cardiovascular health interventions in vulnerable populations has shown that the interventions aimed at decreasing blood pressure are the most promising and behavior change interventions are the most challenging [22,23]. Task-shifting interventions were also proven to be effective in lowering the low-density lipoprotein cholesterol (LDL) and total cholesterol [24]. In another review, significant reduction in the mean blood pressure and glycated haemoglobin levels were seen in task-shifting interventions for the CVD risk reduction in LMICs [24,25].

Guidelines on smoking cessation programs have shown that the combinations of behavioral change and pharmacological support given by the professional healthcare providers are the most effective. CHWs are crucial as they serve as a critical link in increasing the communities' access to services, especially for people living in rural and undeserved areas, which ultimately form an integral part needed to achieve the SDG goals. Therefore, the aim of this study is to evaluate the effectiveness of involving CHWs in smoking cessation programs as compared to the usual care.

## 2. Materials and methods

This review was conducted and reported in accordance to the Preferred Reporting Items for Systematic Reviews and Meta-Analyses (PRISMA) guidelines and based on the Cochrane Collaboration approach [26,27].

### 2.1 Eligibility criteria

We included intervention studies or controlled clinical trials that compared between a combination of interventions given by CHWs and usual care, and only usual care. Only interventional studies conducted among smokers aged 18 years old and above, having the smoking abstinence rate as the outcome were included. We excluded studies that conducted intervention in adolescent or special group (defines as mental illness or LGBTQ). Non-English language studies, reviews, proceedings, qualitative studies, descriptive studies and protocol were also excluded.

## 2.2 Data sources and search strategy

We systematically searched for relevant articles published in The Cochrane Central Register of Controlled Trials (CENTRAL) and PubMed/MEDLINE up to September 2019. The search was limited to 10 years, from 2009 until 2019. We combined the keywords for smoking, cessation and community as the following: cessation or reduction AND smoking or tobacco AND volunteer or peer or community or lay.

## 2.3 Study selection

A pair of authors independently assessed the titles and abstracts of a defined set of articles. Each study was recorded as include, exclude or unclear. The full articles were retrieved for further assessment if they were recorded as include or unclear. Eligible studies were identified based on the inclusion criteria. Any discrepancies in the assessment were resolved by discussion leading to a consensus.

## 2.4 Data extraction, data analysis and risk of bias assessment

Data extraction from all potential studies was documented in a table. The table included information on study characteristics (sample size and study duration), participant characteristics (setting, population type, and specific ethnicity), intervention characteristics (type of interventions given in both arms, and behavioural or pharmacological intervention), training of CHWs (background, duration of training and training module) and analysis and results (outcomes) of both arms. All authors independently extracted the data and any discrepancies were resolved by discussion. The characteristics of the included studies are outlined in Tables 1–3.

Data synthesis and analysis were carried out using Review Manager Software (Rev Man) version 5.3 (Nordic Cochrane Centre, Cochrane Collaboration, Copenhagen). The end of study values were taken. If studies measured smoking cessation rates at multiple intervals (3 or 6 or 9 months), the outcomes in the final point of interval were taken. If studies reported two types of smoking cessation measurements (self-reported or chemically verified using carbon monoxide (CO) level), the chemically verified outcomes were taken.

We assessed the study quality of each study using the COCHRANE guideline for assessment of systematic reviews and the published assessment guide on risks of bias assessment for intervention studies [26]. The studies were evaluated based on eight criteria: randomized treatment order, allocation concealment, blinding of participants and personnel, blinding of outcome assessment, incomplete outcome data, selective outcome reporting and other bias. For each item, risk of bias was classified as 'low risk', 'high risk, or 'unclear risk', with the last category indicating either a lack of information or uncertainty over the potential for bias. The results were presented in a 'Risk of bias' summary (Fig 3a and 3b) and S1 Table.

# 3. Result

## 3.1 Search results

We identified 2794 articles through our electronic database search. After excluding 280 duplicated studies, a total of 2614 articles were excluded following titles and abstracts screening due to irrelevant study designs (observational studies or interventional studies without control group), irrelevant outcomes (no smoking cessation or did not provide smoking abstinence rate), and irrelevant interventions (not given by CHWs). One hundred full text articles were screened, of which 95 articles were excluded for following reasons: interventions were not given by CHWs (73 studies), study designs were non controlled trials (17 studies) and

**Table 1. Characteristics of the studies.**

| Study | Setting | Participants type | Sample size | Intervention | Control | Mean cigarettes/day | Study Design | Study Duration |
|-------|---------|-------------------|-------------|--------------|---------|---------------------|--------------|----------------|
| **Bernstein et al, 2011** [28] | Bronx, New York, United States of America | Hispanic and African American | 338 | 170 | 168 | 15 ± 7.48 | RCT | 21 months |
| **Wang et al, 2017** [30] | Hong Kong | Public | 1,226 | 402 | 416 408 | N/A | Cluster-3 arms RCT | 3 months |
| **White et al, 2018** [29] | Maryland, United States of America | Public | 200 | 101 | 99 | 19.3 ± 18.3 | RCT | 6–8 weeks |
| **Bonevski et al, 2018** [32] | New South Wales, Australia | Disadvantage Adult clients of the Community Care Centre | 431 | 187 | 244 | 15 ± 7.46 | Parallel randomised trial | 30 months |
| **Jiang et al, 2018** [31] | Thai Nguyen, Vietnam | Village Population | 1,318 | 781 | 537 | 11.02 ± 9.48 | Quasi-experimental | 6 months |

Abbreviations: RCT- randomized controlled trial.

outcomes were not smoking cessation (5 studies). Finally, five studies were included in this review [28–32]. Fig 1 shows PRISMA flowchart.

## 3.2 Characteristics of included studies

A total of 3513 participants were included in this review. The studies were conducted between the year of 2011 and 2018. Two studies were conducted in United States of America (USA) [28,29] while the other studies were conducted in Hong Kong [30], Australia [32] and Vietnam [31]. The duration of the studies conducted ranged from 6 weeks to 30 months. All studies were conducted among the community, except for one study conducted in an emergency department setting [28]. The same study was conducted on a specific ethnicities (African American and Hispanic community) [28]. There were no significant differences between the baseline number of cigarettes per day (13.08 ± 9.57). All studies were randomized controlled trial studies except for study by Jiang et al. [31].

**Table 2. Characteristic of intervention and outcome.**

| Study | Intervention | Control | Outcome Intervention | Outcome Control |
|-------|--------------|---------|----------------------|-----------------|
| **Bernstein et al, 2011** [28] | Smoking cessation brochure (10–15 min) AND MI by interventionist, 6 weeks course of NRT. | Smoking cessation brochure and contact information to smoking cessation programs | Self-reported 7-days PPA rates at 3 months: 14.7% | Self-reported 7-days PPA rates at 3months: 13.2% |
| **Wang et al, 2017** [30] | Brief advice using structural model AWARD, health warning leaflet AND active referral to SC services by ambassadors | Brief advice using structural model AWARD, health-warning leaflet and encourage to SC services | Validated abstinence rates at 6 months: 9.0% | Validated abstinence rates at 6 months: 5.0% and 5.1% |
| **White et al, 2018** [29] | Automated SFTXT AND personalized text messages from peer mentor | Automated SFTXT messages | Biochemically verified PPA rates at 3 months: 7.9% | Biochemically verified PPA rates at 3 months: 3.0% |
| **Bonevski et al, 2018** [32] | On screen advice to quit smoking, state Quitline telephone number, and a gift bag with call it quits AND MI and NRT by trainer volunteer case workers | On screen advice to quit smoking, state Quitline telephone number, and a gift bag with call it quits | Continuous verified PPA at 6 months: 1.0% | Continuous verified PPA at 6 months: 1.4% |
| **Jiang et al, 2018** [31] | 4As (brief counselling and educational materials) AND refer smokers to a trained VHW (4As+R) | 4As (brief counselling and educational materials) | Validated abstinence rate at 6 months: 25.7% | Validated abstinence rate at 6 months: 10.5% |

Abbreviations: MI- Motivational Interviewing; NRT- nicotine replacement therapy; PPA- point prevalence abstinence; SC- smoking cessation; CHW-Community Health Worker; AWARD; SFTXT- SmokefreeTXT; SC- smoking cessation; 4As- Ask (screen for tobacco use), Advise to quit, Assess readiness to quit, and Assist; VHW-Village Health Worker.

**Table 3.** Characteristic of community health worker and training of community health worker.

| Study | Amount and Description | Educational Background | Experience | Incentive | Duration/ Training Module/ Supervision |
|---|---|---|---|---|---|
| **Bernstein et al, 2011** [28] | One (Interventionist) | Not reported | Not reported | None | Duration: 2 weeks |
| | Former smoker | | | | Training module: Course and practicum (role play, slide shows and observation)- Epidemiology, health effects and treatment of tobacco dependence, motivational interviewing. |
| | ED based | | | | Supervision: Every two weeks |
| **Wang et al, 2017** [30] | Not mentioned (SC ambassador) | University students (health-related studies)* | Volunteers from NGO*. | None | Duration: 4 hours |
| | | | | | Training module: Tobacco control and SC, SC reduction advice skills |
| | | | | | Supervision: Spot checks |
| **White et al, 2018** [29] | Thirty-two (Peer mentor) | Not reported | Facilitators of the American Cancer Society Freshstart group-based cessation support program*. | $200 and entry into $1000 drawing | Duration: 2 hours |
| | Former smoker more than 1 year | | | | Training module: Online- study details, smoking and SC, MI, web-based-text-messaging platform |
| | 18 years old and above | | | | Supervision: N/A |
| | Lived in the United States | | | | |
| | Willingness to mentor smokers through texting and completing the online training program | | | | |
| **Bonevski et al, 2018** [32] | Not mentioned (Caseworker) | Not reported | Volunteer case workers of community social service organizations (NGO) in NSW. | None | Duration: One day |
| | | | | | Training module: Behavioral counselling and MI |
| | | | | | Supervision: N/A |
| **Jiang et al, 2018** [31] | Six to eight (VHW) | Not reported | Worked as a VHW at that particular site for a year or more. | None | Duration: 4 days |
| | Not a current smoker and willing to participate in the required components of the study intervention. | | | | Training module: Research ethics, study protocol, use of CO monitor, theory of behavior change, MI, social cognitive skills building approach. |
| | | | | | Supervision: N/A |

Abbreviations: N/A- not available; SC-Smoking Cessation; MI-motivational Interviewing; ED- Emergency Department; NGO-non-governmental organizations; NSW-New South Wales; VHW-village health worker.

*Details of the experience needed were not available.

## 3.3 Characteristic of intervention

All of the studies used behavioral intervention such as brief advice as the method for smoking cessation. The usual care intervention given to both intervention groups and control group were mainly brief advice, and some of the studies provided self-help material. Four studies used brief advice; one study used AWARD model (Ask, Warn, Advise, Refer and Do-it-again) [30], one study used 4A's model (Ask for tobacco use, Advise to quit, Assess readiness to quit, and Assist) [31], one study used automated Smokefree TXT [29] and one study used on-screen advice [32]. Self-help materials in the form of leaflet or brochure were provided to the participants in three studies [28,30,31].

The interventions given by CHWs in the studies were behavioral intervention, active referral and pharmacological therapy. Two studies used motivational interviewing (MI), and provided pharmacological treatment (nicotine replacement therapy (NRT)) to the participants [28,32]. One study had active referral to the existing smoking cessation intervention in clinics

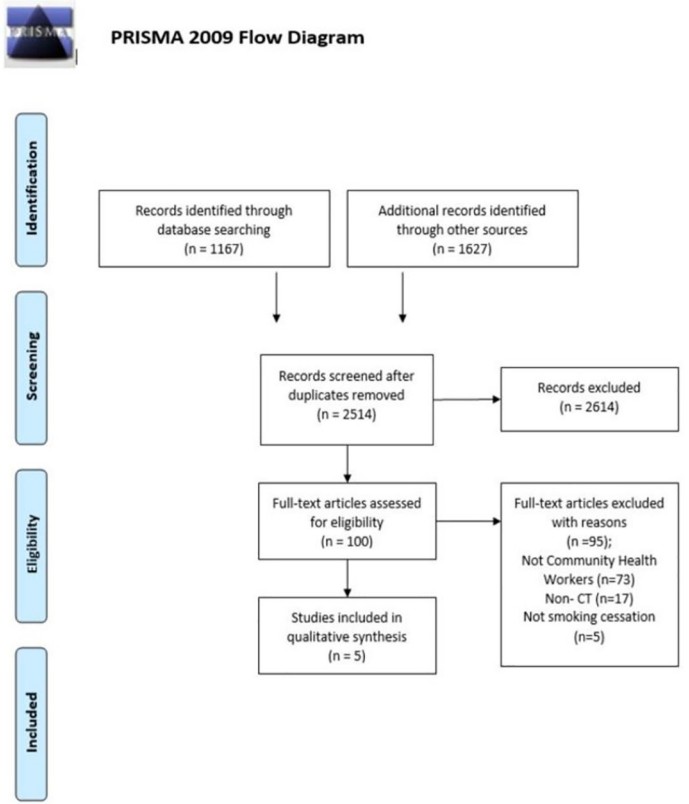

**Fig 1. PRISMA flowchart for the selection of studies.** Outcomes of the systematic review of the literature by record identification, screening, and analysis in the Preferred Reporting Items for Systematic Reviews and Meta-Analyses (PRISMA) statement flow diagram.

by CHWs [30]. One study used personalized text messages [29] as a mediator for smoking cessation intervention. Table 2 shows the details of the interventions.

### 3.4 Smoking abstinence rate

All studies used intention to treat analysis, except for the study by Bonevski et al. [32]. The outcomes of the smoking abstinence rate were measured by either self-reporting or chemically verified using CO level. Two studies provided abstinence rate after 3 months [28,29] and the remaining three studies after 6 months [30–32].

The pooled OR were estimated using random effects models as the studies included were heterogenous with respect to interventions and populations. Overall, all studies had higher odds of abstinence in the intervention group as compared to the usual group [OR 1.95, 95% CI (1.35,2.83)]. Two studies reported significant smoking abstinence rate in the interventions using CHWs, with OR 2.98 [2.16, 4.10] [31], OR 1.81 [1.04, 3.16] (active referral vs brief advice group), and OR 1.85 [1.06, 3.23] (active referral vs control group) [30]. Fig 2 and Table 2 show the details of the result.

### 3.5 Characteristics and training of community health workers (CHWs)

The number of CHWs in the study varied from one person to 32 persons; two studies did not provide the amount of CHWs [30,32]. Only one study had CHWs from the same community with the participants [31]. Two studies had volunteers from NGOs [30,32] or university

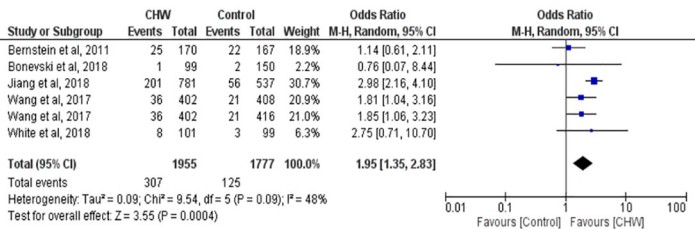

**Fig 2. Forest plot shows smoking abstinence rate.**

students [30]. CHWs from two studies had prior experiences working as CHWs or worked on smoking cessation programs previously [29,31]. Two studies had CHWs who were former smokers [28,29].

The duration of training of the CHWs ranged from just two hours to two weeks. The training were given by either the researchers or experts in the field. The content of the training was comprised of the study protocol and method in smoking cessation intervention. Supervisions were given weekly or biweekly to ensure fidelity. Only one study provided incentives to the CHWs [29]. Table 3 shows the details of the characteristics of CHWs and their training.

## 3.6 Risk of bias assessment

We judged the risk of possible bias present in the studies according to the four incorporated criteria. We presented the summary of risk of bias assessment of the studies in S1 Table and Fig 3a to 3b. Study by Jiang et al. was the only study assessed as high risk in random sequence generation, allocation concealment and blinding as this study was a quasi-experiment study. The random sequence generation was assessed to be of low risk of bias in other studies. When assessing the allocation concealment of the included studies, other studies had low risk of bias except for study by White et al. [33] as they did not clearly state the method of allocating the treatment groups. Performance bias was assessed to be low risk in three studies [28,30,32] with the remaining studies assessed to be high risk. The detection bias was assessed to be low risk in three studies [29,30,32]. Most of the studies had high risk in attrition bias [29,30,32] due to the high dropout rate (>20%) [34]. Three studies with available protocol and all the pre-specified outcomes measured were considered as low risk of reporting bias [29,30,32]. Three studies that measured verified outcome were assessed to be of low risk of other bias [30–32].

## 4. Discussion

This review evaluated the effectiveness of involving CHWs in smoking cessation programs as compared to the usual care. Our review found higher smoking cessation rates in the interventions involving CHWs as compared to usual care in five studies. The quality of the studies were low particularly in attrition bias as three studies reported high dropout rate (20%). Evidence of effectiveness of CHWs in the cardiovascular diseases (CVD) management and prevention is still lacking. Previous studies reported evidences on the four main risk of CVDs: hypertension, diabetes mellitus, unhealthy diet and alcohol and tobacco consumption. A study by Jeet et al. reported an increase in tobacco cessation in interventions using CHWs as compared with the standard care [35], as with this review.

The objective of having CHWs is having the intervention given by someone from the community itself. Its effectiveness vary depending on their training program, demographics and settings [36]. Significant smoking cessation rates were seen in two studies that conducted the

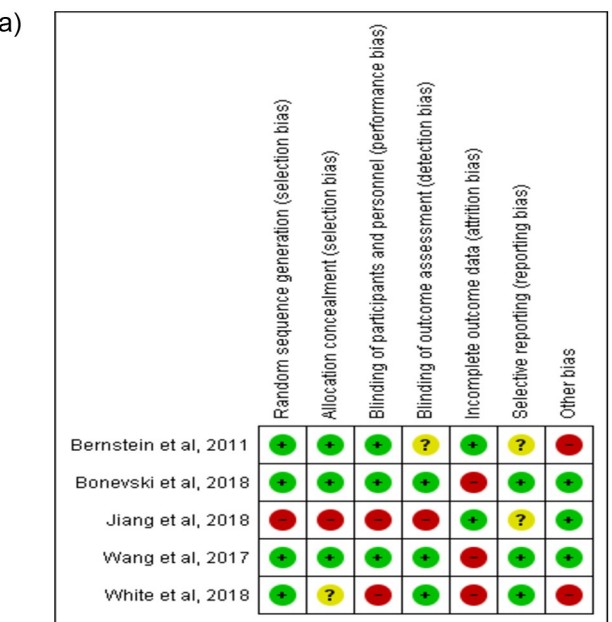

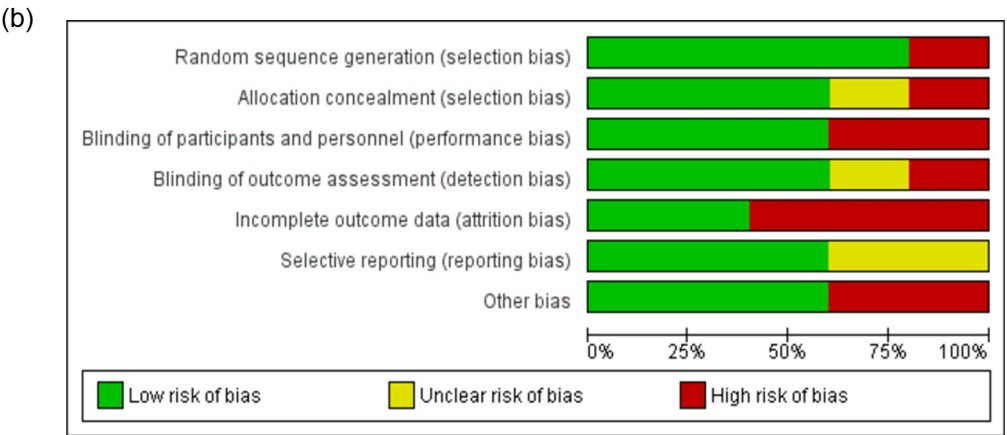

**Fig 3. (a) and (b): Risk of bias assessment.**

training in 4 hours [30] and 4 days [31]. Only one study had CHWs from the same community with the participants, and reported significant reduction of smoking cessation rate. In addition, the study only takes CHWs that have prior experience working as CHWs at that particular site. The performance of CHWs is also associated with their age, skill and educational level [37]. Another study that reported significantly higher cessation rate had university students or volunteers from NGOs as the CHWs [30].

The WHO has formulated a set of recommendations that provides guidance for the task-shifting approaches. These recommendations have implications for a range of health services including the management of NCDs. The main enablers of CHWs interventions are provision of algorithms and protocols, while, restrictions on prescribing medications and availability of medicines are the main barriers identified. Therefore, to ensure its effectiveness, a sound protocol should be developed. However, there was a high heterogeneity in the interventions given

among the included studies. The guidelines on smoking cessation programs showed that a combination of behavioral change and pharmacological support are the most effective. All of the studies used behavioural intervention as the method for smoking cessation with two studies providing pharmacological treatment (nicotine replacement therapy) to the participants. Our review showed that significant higher abstinence rates were seen in one study that used active referral, AWARD and leaflet [30] and one study that used 4A's [31] as the method for behavioural intervention.

CHWs productivity is also largely determined by the condition under which they work. The focus on the provision of an enabling work environment for CHWs is essential for achieving high levels of productivity. Jaskiewicz et al. presented a model in which the work environment encompassed four essential elements: workload, supportive supervision, supplies and equipment, and respect from the community and the health system [38]. Similarly, important determinant of positive CHWs interventions is community embeddedness, meaning that community members have a sense of ownership of the program [39].

There are several limitations in this review. First is the high heterogeneity in baseline characteristics of the participants and study designs. Theoretical health belief models assert that a person who is in contemplation stage may be easily influenced to receive intervention. Therefore, the studies with the participants of low exposure to smoking and of intention to quit (theoretical health belief models) might be attributed to higher cessation rates. One study that selected participants who already had the intention to quit smoking showed similar results [29]. Similarly, one study that included participants with low amount of cigarettes used (one in the past 3 months or 1 months) showed significant smoking cessation rates [30]. Meanwhile, one study conducted in a hospital setting (emergency department) reported higher abstinence rate in smokers who had smoking-related health issues [28]. The review of 33 studies reported high-certainty evidence that incentives improve smoking cessation rates [40]. Our review showed similar result, whereby one study that provided incentives to the participants had significant smoking cessation rates [30].

Higher retention is significantly associated with higher quit rates. However, there were high attrition rates (loss of follow up of >20%) in most of the studies included in this review, with the highest attrition rates of 42% [32]. The reviews reported that increasing age, higher level of education and higher motivation to quit were associated with higher retention [41].

## 5. Recommendations

Previous review has reported cost-effectiveness of CHWs interventions as compared to standard care particularly in tuberculosis, malaria and maternal and child health [42,43]. We recommend future studies to analyse the cost-effectiveness of using CHWs in smoking cessation programs. Since the effectiveness of CHWs intervention is largely dependent on the framework and training module, it is imperative that the guidelines on the framework are drawn up. We also recommend studies to measure abstinence rates of smoking according to the gold standard.

## 6. Conclusion

CHWs have the potential to bridge between primary healthcare providers and communities, and consequently reducing the gap, especially among low socioeconomic populations as recommended by the SDG. Our review reported positive outcomes on using a combination of CHWs and usual care in smoking cessation as compared with usual care alone, however the evidences are insufficient and have high heterogeneity.

## Supporting information

**S1 Checklist. PRISMA checklist.**
(DOC)

**S1 Table. Risk of bias details.**
(DOCX)

## Acknowledgments

We would like to thank the Director General of Health Malaysia, Director of Institute for Medical Research for the permission to publish this article.

## Author Contributions

**Conceptualization:** Siti Hafizah Zulkiply, Rosliza Abdul Manaf.

**Data curation:** Siti Hafizah Zulkiply, Lina Farhana Ramli, Zul Aizat Mohamad Fisal, Bushra Tabassum.

**Formal analysis:** Siti Hafizah Zulkiply.

**Methodology:** Siti Hafizah Zulkiply.

**Writing – original draft:** Siti Hafizah Zulkiply.

**Writing – review & editing:** Siti Hafizah Zulkiply, Rosliza Abdul Manaf.

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
