## [Decision Letter · Decision Letter 0]

24 Jun 2020

PONE-D-20-06267

Effectiveness of Community Health Workers in Smoking Cessation Programme: A Systematic Review

PLOS ONE

Dear Dr. zulkiply,

Thank you for submitting your manuscript to PLOS ONE. After careful consideration, we feel that it has merit but does not fully meet PLOS ONE’s publication criteria as it currently stands. Therefore, we invite you to submit a revised version of the manuscript that addresses the points raised during the review process.

We look forward to receiving your revised manuscript.

Kind regards,

Stanton A. Glantz

Academic Editor

PLOS ONE

Journal Requirements:

4. Please ensure that you refer to Figure 2 in your text as, if accepted, production will need this reference to link the reader to the figure.

5. Please include a copy of Table 4 which you refer to in your text on page 17.

Additional Editor Comments (if provided):

Reviewers' comments:

Reviewer's Responses to Questions

**Comments to the Author**

1. Is the manuscript technically sound, and do the data support the conclusions?

Reviewer #1: Partly

Reviewer #2: Partly

2. Has the statistical analysis been performed appropriately and rigorously? 

Reviewer #1: N/A

Reviewer #2: N/A

3. Have the authors made all data underlying the findings in their manuscript fully available?

Reviewer #1: Yes

Reviewer #2: No

4. Is the manuscript presented in an intelligible fashion and written in standard English?

Reviewer #1: Yes

Reviewer #2: No

5. Review Comments to the Author

Reviewer #1: Thank you for the opportunity to review this manuscript. CHWs and task shifting are relevant to be considered in interventions for smoking cessation.

Three of the main points that I would like to mention about the paper are:

- Results and conclusions: It needs to describe, more in depth, the results and discussion that supports the objective of the paper. Taking into account the main outcomes that you are studying. In terms of outcomes you can include significance of the OR, or more details.

- Results: Figure 2 is not described in the text and also not mentioned in methods. Please include some information about it.

- The paper presents some grammatical errors and some sentences that need to be clarified.

Please find additional questions/comments below:

Material and Methods:

- Was the review registered in PROSPERO?

- Can you provide more specifications about what type of studies were included? e.g. clinical trials, clinical answers, etc

- There is no information about the duplicated data and its exclusion. Was duplicated information founded? If so, the information should also be shown in the PRISMA flow diagram (Results)

Data Extraction, Data Analysis and Risk of Bias assessment:

- In terms of effectiveness, more details of the analysis and results (outcomes) should be described

Conclusions:

- Describe more about the limitations of the review in terms of participants and study designs and also availability of data.

References: In general the consistency and format of references should be revised: pages, p at the ending, links, etc

- Review first reference, pages missing

- Reference 6: review the reference and format

- Line 44-46: Reference missing

- Line 55: Is reference 3 correct?

- Line 66: "forecasted that more than 8 million people will die from diseases related to tobacco use by year 2030 if pattern of smoking continues" is the reference the same?

- Line 72-74: "with the highest prevalence of smokers were among 25 to 44 years old age group (28%), followed by 45 to 64 years old age group (20%) (3)." Are these group ages of specific special interest? Because in the non aggregated data of the original paper there is variation in that age group.

- Line 119-121: Include references

- Line 176: Reference 73 is not available

Please review the following typos, minor grammatical errors, that should be corrected:

Line 2 check punctuation of the title

Line 14: by one-third at the end

Line 18: Has shown “in the prevention”

Line 19: this systematic review is aimed //// evidence (not plural)

Line 28: where only 5

Line 32: Review consistency with the abbreviation CHWs (instead of CHWS)

Line 44: NCD are the

Line 49: Reduce by one third (same as summary)

Line 67: die due to tobacco. There were 933.1

Line 70: the prevalence …was

Line 73: being among the 25 to 44 age group, followed by the 45 to 64 (20%)

Line 76: programs

Line 77: primary care settings

Line 77-79: Interventions: What do you refer in this paragraphs?

Line 79: Since 2004, Malasia has… clinics… that are available at most primary health clinics.

Line 81: Areas

Line 82: Because LSS is an important determinant of smoking, it is…. cessation. (exclude in this area)

Line 97 to 99: keep the first should and the others are redundant (you just need a comma)

Line 111: programs showed that a combinations….

Line 112: CHWs are crucial

Line 113: serve as a critical link124: compared a combination

Line 124 -127: Eligibility criteria: we included is redundant, is mentioned 4 times in 4 lines.

Line 131: sources

Line 134: was instead of were

Line 134: we combined (suggestion to keep we just at the beginning_)

Line 139: what were the defined set of articles, does it mean that was independently for all of them?

Line 149: Include the term of both arms

Line 172: electronic database search

Line 180: Review the characters for subtitles and figures

Line 189: among the community…. in an emergency department setting

Line 190: One study was target on…

Line 192: the number of …

Line 235: 32 people, (exclude with)

Line 239: who were instead of “whom was”

Line 240: range instead of “ranging”

Line 241: training was given

Line 242: content of the training was comprised….

Line 243: Supervisions were given weekly or biweekly [this lines sound incorrect]

Line 262-268: Performance …. Rate(>20%) [this paragraph is not clear, please review it while it contains editing errors]

Line 274: There is considerable evidence supporting the impact…

Line 279: Task shifting interventions were proven to be

Line 284: our review has also found similar findings effective

Line 308: the smoking cessation programme showed that a combination

Line 310: two studies providing

Line 325: The study? One study?

Line 328: had experience

Line 341: on the framework… therefore, it is pivotal that guidelines on the framework are drawn up.

Line 342: studies measure

Line 348: reported a positive outcome or reported positive outcomes … on using a combination

Line 353: We would also like to…

Reviewer #2: Thank you for writing an important paper on CHWs and their potential to impact smoking cessation programs amongst other health promotion activities. The authors put forward a good case in their background however paper lacks methodological rigour and needs additional information in the results. Discussion doesn't reflex the findings. Please re-write and I would consider re-reviewing. I have included detailed comments on the manuscript. I am unable to upload so will send via email. Thank you. Best wishes.

6. PLOS authors have the option to publish the peer review history of their article (what does this mean?). If published, this will include your full peer review and any attached files.

Reviewer #1: No

Reviewer #2: No

---

## [Author Response · Author response to Decision Letter 0]

6 Sep 2020

Reviewer 1

Results: Figure 2 is not described in the text and also not mentioned in methods. Please include some information about it.

 Figure 2 is described in line 226 under heading 3.4 Smoking abstinence rate 

Was the review registered in PROSPERO? 

 The review was not registered in PROSPERO

Can you provide more specifications about what type of studies were included? e.g. clinical trials, clinical answers, etc 

 Type of studies included is describe in line 125 under eligibility criteria- This study only included intervention studies or controlled clinical trials that compared a combination of interventions given by CHWs and usual care with only usual care. 

There is no information about the duplicated data and its exclusion. Was duplicated information founded? If so, the information should also be shown in the PRISMA flow diagram (Results)

 Data of duplicates removed is shown in PRISMA flowchart (Figure1) and described in 3.1 search result sections

Data Extraction, Data Analysis and Risk of Bias assessment:

- In terms of effectiveness, more details of the analysis and results (outcomes) should be described

 OR of the studies has been included 

Conclusions:

- Describe more about the limitations of the review in terms of participants and study designs and also availability of data.

 Limitation on the heterogeneity and insufficient study are included 

References: In general, the consistency and format of references should be revised: pages, p at the ending, links, etc

Review first reference, pages missing

 Reference 1 has been revised 

Reference 6: review the reference and format

 Reference 6 has been revised

Line 44-46: Reference missing 

 Reference added for line 44-46

Line 55: Is reference 3 correct?

 The reference has been updated 

Line 66: "forecasted that more than 8 million people will die from diseases related to tobacco use by year 2030 if pattern of smoking continues" is the reference the same?

 Yes, the reference is the same

Line 72-74: "with the highest prevalence of smokers were among 25 to 44 years old age group (28%), followed by 45 to 64 years old age group (20%) (3)." Are these group ages of specific special interest? Because in the non-aggregated data of the original paper there is variation in that age group.

 No, its not of specific interest

Line 119-121: Include references

 Reference included 

Line 176: Reference 73 is not available 

 73 was referred to number of studies and not references.

 

Reviewer 2

Need more data on prevalence of smokers in rural and urban is included

 95% CI is included

Add details on search for publication years

 Details on search for publication years from 2009 to 2019, was added in line 137

Details on population type

 Added specific ethnicity in line 151

Details on type of intervention

 Added behavioural or pharmacological in line 152

Details on validated abstinence rate

 Added chemically verified using CO in line 163

Details on irrelevant study designs 

 Added observational studied or interventional studies without control group in line 178

Details about CHWs, differences by demographics 

 Added level of education in table 3, discuss differences by demographics 

Details on differences of targeted communities with cultural backgrounds

 Details on differences of targeted communities included

Details on behavioral interventions

 Details on behavioral interventions included 

Risk of bias 

 The risk of bias assessment was done according to COCHRANE guideline for assessment of systematic reviews. The details of the risk of bias assessment is presented in S2 Table

---

## [Decision Letter · Decision Letter 1]

8 Oct 2020

PONE-D-20-06267R1

Effectiveness of Community Health Workers Involvement in Smoking Cessation Programme: A Systematic Review

PLOS ONE

Dear Dr. Zulkiply,

Thank you for submitting your manuscript to PLOS ONE. After careful consideration, we feel that it has merit but does not fully meet PLOS ONE’s publication criteria as it currently stands. Therefore, we invite you to submit a revised version of the manuscript that addresses the points raised during the review process.

Reviewer 1 has recommended that this paper be rejected, but I am willing to give you another chance to respond with appropriate revisions.  Reviewer 2 also has some suggestions in an attached file.

We look forward to receiving your revised manuscript.

Kind regards,

Stanton A. Glantz

Academic Editor

PLOS ONE

Reviewers' comments:

Reviewer's Responses to Questions

**Comments to the Author**

1. If the authors have adequately addressed your comments raised in a previous round of review and you feel that this manuscript is now acceptable for publication, you may indicate that here to bypass the “Comments to the Author” section, enter your conflict of interest statement in the “Confidential to Editor” section, and submit your "Accept" recommendation.

Reviewer #1: (No Response)

Reviewer #2: All comments have been addressed

2. Is the manuscript technically sound, and do the data support the conclusions?

Reviewer #1: Partly

Reviewer #2: Yes

3. Has the statistical analysis been performed appropriately and rigorously? 

Reviewer #1: I Don't Know

Reviewer #2: N/A

4. Have the authors made all data underlying the findings in their manuscript fully available?

Reviewer #1: (No Response)

Reviewer #2: Yes

5. Is the manuscript presented in an intelligible fashion and written in standard English?

Reviewer #1: No

Reviewer #2: Yes

6. Review Comments to the Author

Reviewer #1: The paper still contains editing mistakes, please have a further review. For example:

Line 49 'with goal the to “ensure”’

Line 57 'to reduce 30% relative reduction’

Line 267 roleplay

The following comments are related to the tables, please review your tables of results and adjust all the necessary information:

Bernstean et all has 168 patients in the control group, but the table 1 reported 167, the values reported in the study for cigarettes/day were median and IQR for the sample of 168, but the ones reported in the table are Mean and SD for 167.

Table S2 reports 26% of incomplete outcome data, was this related with the 26+31 missing information and/or the expired information over the total sample of the randomized data (338)? if so, the estimation is not correct.

Please verify the results of the OR reported data. In table 2, Bonevski et al, 2018, we observe that the OR from the Self-reported continuous verified PPA at 6 months is 0.76, while in the paper the "continuous self-reported PPA at 6 months” is 1.95 and the OR from the "Continuous verified PPA at 6 months" is 0.77.

Please review the results and forest plot

Table 2, reference 29, the Biochemically verified abstinence at 3 months is 8/101 in the intervention group and 3/99 in the control group. The self reported abstinence at 3 months was 24/101 in the intervention and 13/99 in the control group. If you consider to add both, please include the information.

Figure 2. Taking into account that the Jiang et al study presents 25.7% of abstinences rate, should the number of events in the CHW group be 201 instead of 200?

Reviewer #2: (No Response)

7. PLOS authors have the option to publish the peer review history of their article (what does this mean?). If published, this will include your full peer review and any attached files.

Reviewer #1: No

Reviewer #2: No

---

## [Author Response · Author response to Decision Letter 1]

17 Oct 2020

Dear Reviewers, 

Thank you for your kind and deliberate comments. Please find below the responses to your comments. 

Reviewer #1: The paper still contains editing mistakes, please have a further review. For example:

Line 49 'with goal the to “ensure”’

Line 49 has been edited 

Line 57 'to reduce 30% relative reduction’

Line 57 has been edited 

Line 267 roleplay

Line 267 has been edited 

The following comments are related to the tables, please review your tables of results and adjust all the necessary information:

Bernstein et all has 168 patients in the control group, but the table 1 reported 167, the values reported in the study for cigarettes/day were median and IQR for the sample of 168, but the ones reported in the table are Mean and SD for 167.

The n for the control group has been changed to 168 and we have recalculated the mean and sd accordingly. 

Table S2 reports 26% of incomplete outcome data, was this related with the 26+31 missing information and/or the expired information over the total sample of the randomized data (338)? if so, the estimation is not correct.

The percentage of loss to follow up has been recalculated and updated in the Table S2 and Figure 3 (a) and 3(b). 

Please verify the results of the OR reported data. In table 2, Bonevski et al, 2018, we observe that the OR from the Self-reported continuous verified PPA at 6 months is 0.76, while in the paper the "continuous self-reported PPA at 6 months” is 1.95 and the OR from the "Continuous verified PPA at 6 months" is 0.77.

The outcome for Bonevski et al has been changed accordingly. 

Please review the results and forest plot

Table 2, reference 29, the Biochemically verified abstinence at 3 months is 8/101 in the intervention group and 3/99 in the control group. The self-reported abstinence at 3 months was 24/101 in the intervention and 13/99 in the control group. If you consider to add both, please include the information.

We have made the correction and only included the biochemically verified outcome. The forest plot has been adjusted accordingly in Figure 2. 

Figure 2. Taking into account that the Jiang et al study presents 25.7% of abstinences rate, should the number of events in the CHW group be 201 instead of 200?

The number of events for Jiang et al has been corrected.

---

## [Editor Report · Decision Letter 2]

9 Nov 2020

Effectiveness of Community Health Workers Involvement in Smoking Cessation Programme: A Systematic Review

PONE-D-20-06267R2

Dear Dr. Zulkiply,

We’re pleased to inform you that your manuscript has been judged scientifically suitable for publication and will be formally accepted for publication once it meets all outstanding technical requirements.

Kind regards,

Stanton A. Glantz

Academic Editor

PLOS ONE
---

## [Editor Report · Acceptance letter]

11 Nov 2020

PONE-D-20-06267R2 

Effectiveness of Community Health Workers Involvement in Smoking Cessation Programme: A Systematic Review. 

Dear Dr. Zulkiply:

I'm pleased to inform you that your manuscript has been deemed suitable for publication in PLOS ONE. Congratulations! Your manuscript is now with our production department. 

Kind regards, 

on behalf of

Professor Stanton A. Glantz 

Academic Editor

PLOS ONE